# Association of Fortification with Human Milk versus Bovine Milk-Based Fortifiers on Short-Term Outcomes in Preterm Infants—A Meta-Analysis

**DOI:** 10.3390/nu16060910

**Published:** 2024-03-21

**Authors:** Radu Galis, Paula Trif, Diana Mudura, Jan Mazela, Mandy C. Daly, Boris W. Kramer, Shivashankar Diggikar

**Affiliations:** 1Department of Neonatology, Emergency County Hospital Bihor, 50 Clujului Street, 410053 Oradea, Romania; raduoradea@yahoo.co.uk (R.G.);; 2Department of Neonatology, Poznan University of Medical Sciences, 10 Fredry Street, 61-701 Poznan, Poland; 3Doctoral School of Biomedical Sciences, University of Oradea, 1 University Street, 410087 Oradea, Romania; 4Irish Neonatal Health Alliance (INHA), 26 Oak Glen View, Southern Cross, A98 Y234 Bray, Ireland; 5Oyster Woman and Child Hospital, Bengaluru 560043, India; shiv.diggikar@gmail.com

**Keywords:** nutrition, supplement, bovine milk protein, human milk fortifier, extremely preterm infants

## Abstract

This meta-analysis assessed short-term outcomes after using human milk-derived fortifiers (HMFs) compared with bovine milk fortifiers (BMFs) in preterm infants fed an exclusive human milk (HM) diet, either mother’s own milk (MOM) or donor human milk (DHM). We searched PubMed, Embase, Google Scholar, CENTRAL and CINHAL between January 2015 and August 2023 for studies reporting outcomes in infants with ≤28 weeks gestation and/or birthweight ≤ 1500 g on an exclusive human milk diet fortified with HMF versus BMF. The primary outcomes were death and NEC (stage ≥ 2). Four studies with a total of 681 infants were included. Mortality was significantly lower in infants fed with an HM-HMFs diet (four studies, 681 infants; RR = 0.50, 95% CI = 0.26–0.94; *p* = 0.03; *I*^2^ = 0%), NEC was similar between the two groups (four studies, 681 infants; RR = 0.48, 95% CI = 0.20–1.17; *p* = 0.11; *I*^2^= 39%). BPD was higher in the HM-BMFs group (four studies, 663 infants; RR = 0.83, 95% CI = 0.69–1.000; *p* = 0.05, *I*^2^ = 0%), although not statistically significant. No differences were found for sepsis (RR = 0.97, 95% CI = 0.66–1.42; *p* = 0.96; *I*^2^ = 26%) or combined ROP (four studies, 671 infants; RR = 0.64, 95% CI = 0.53–1.07; *p* = 0.28; *I*^2^ = 69%). An HM-HMFs diet could possibly be associated with decreased mortality with no association with NEC, BPD, sepsis, or ROP. This meta-analysis was limited by the small number of studies included. However, the results should not be refuted for this reason as they provide an impetus for subsequent clinical trials to assess the observed associations.

## 1. Introduction

Neonatal care has made significant progress in recent decades. As a result, the survival rate of high-risk infants, including those born with extremely and very-low birth weights, has increased. The future goal is to continue to reduce the mortality rate while also reducing morbidities.

Preterm birth exposes a baby to the environment outside of the uterus with immature organs that have to adapt to this very hostile environment after birth [1]. Premature delivery invariably results in very different neurodevelopment with effects lasting into adulthood [2]. Multiple complications such as necrotizing enterocolitis (NEC), bronchopulmonary dysplasia (BPD), late-onset sepsis (LOS), and retinopathy of prematurity (ROP), are independently associated with adverse neurological outcomes [3,4,5,6,7]. Preterm infants must adjust to extrauterine life without the caloric and nutritional supply from the placenta. It is challenging to achieve postnatal weight gain similar to that in the intra-uterine phase [8]. Taken together, this explains the rationale for optimizing nutrition and nutrients to reduce the long-term risk of neurodevelopmental impairment (NDI) [9]. 

Mother’s own milk (MOM) has been proven and recognized to be the best choice, especially for preterm infants [9]. Pasteurized pooled human donor milk is supplemented if the MOM is insufficient or not available. However, pasteurized pooled human donor milk, and a baby’s own mother’s milk are very different in terms of both nutritional and non-nutritional components, which is rarely recorded and reported in clinical trials. A position paper from ESPGHAN on enteral nutrition in preterm infants stated that MOM alone does not meet the high nutritional requirements of the rapidly growing preterm infant, which mandates fortification [10,11]. Data on nutrition accretion support the use of bovine fortifiers or human milk fortifiers but data from large clinical trials are lacking. The traditional practice is to supplement the MOM or donor milk with fortifiers derived from bovine milk (BMFs). This practice introduces bovine milk protein into the enteral therapy from fortification onwards. Formula feeding with bovine milk products have been associated with an increased rate of NEC depending on the time of their introduction into enteral therapy [12,13], as well as sepsis and death. Fortifiers derived from donated human milk (HMFs) make an exclusive human milk diet possible. The clinical benefits are not completely clear and the costs of an HMFs diet are extremely high. There is a need for a systematic review and meta-analysis assessing whether an exclusive human milk diet supplementing human milk with HMFs (HM-HMFs) has beneficial impacts on the risk of death, NEC, BPD, LOS, or ROP when compared to human milk supplemented with BMFs (HM-BMFs).

## 2. Material and Methods

### 2.1. Protocol and Registration

This systematic review and meta-analysis is registered with PROSPERO International prospective register of systematic review (CRD42023466837).

### 2.2. Eligibility Criteria

This review and meta-analysis considered experimental and observational studies and subgroup analyses of sufficient size. 

Infants born after ≤28 weeks of gestation and/or with birthweight ≤ 1500 g were considered for inclusion. Studies comparing the effects of an exclusive human milk diet with human milk-derived fortifiers to the exposure to human milk fortified with bovine milk-derived fortifiers were included.Outcomes included NEC, sepsis, or LOS, BPD, ROP, death, feeding intolerance, and growth velocity.

### 2.3. Information Sources and Search Strategy 

Electronic database searches were conducted in PubMed, Google Scholar, CINAHL Plus with Full Text (EBSCOhost, Ipswich, MA, USA), Embase (Elsevier, Amsterdam, The Netherlands), and the Cochrane Central Register of Controlled Trials (CENTRAL, Wiley, Hoboken, NJ, USA) in English language. The search strategy used keywords like preterm infants, human milk, human milk-derived fortifiers, and outcomes relevant to this review. Searches were carried out by DM, PT, RG, and BK. Full texts of articles that met the eligibility criteria by title and abstract were retrieved and screened by two authors for inclusion, including the references. Disagreements were resolved by discussion between the authors. The trial authors were contacted by email to request missing data. Details of the search strategy are in the online Appendix A.

### 2.4. Study Selection

Randomized or quasi-randomized clinical trials, observational studies, and subgroup analyses from clinical trials that evaluated at least one of the outcomes were included. Preterm infants born after ≤28 + 0 weeks gestation and/or with a birthweight ≤1500 g were included in the study if their feeds were based on human breast milk fortified either with a human milk-based fortifier (HM-HMF) or a bovine milk-derived fortifier (HM-BMF). Information on MOM or pooled DM as basis of the human milk diet was recorded if available. Infants who had been exposed to bovine milk before 34 weeks were excluded.

A total of 1449 articles were identified through database searching. After duplicates were removed, 1107 articles underwent title and abstract screening. Twenty-two full-text articles were assessed. A final total of 4 articles including 3 RCTs [14,15,16], and 1 non-randomized study of interventions (NRSIs) were deemed eligible to be included in the final meta-analysis (Table 1). 

The PRISMA flow diagram is shown in Figure 1.

### 2.5. Definitions and Measurements

NEC

In the OptiMoM, Sullivan and Jensen trials, the diagnosis of NEC was made by radiologists, who were blind to the dietary assignment to improve diagnostic reliability. Final assessment of NEC diagnosis was performed by a blinded consensus panel review in the Jensen study [14].

BPD

All included studies [14,15,16,18] used BPD as need for oxygen requirement at 36 weeks postmenstrual age.

Sepsis/LOS

Jensen et al. reported data on sepsis [14]. The other three studies reported late onset sepsis as the outcome [15,16,18]. LOS was defined as a positive culture in blood, cerebrospinal fluid, or suprapubic or catheter urine ≥ 5 days after birth in newborns with clinical deterioration and a laboratory inflammatory response.

ROP

Two studies examined severe ROP [15,16], and one examined all ROP [18]. ROP was diagnosed in the Jensen study after postmenstrual week 42 + 0 according to international classification into stages I–V [14]. We opted to combine the four studies despite heterogeneity.

Feeding intolerance

Both in the Jensen study [14] and in the OptiMom [15] study feeding intolerance was quantitated by the number of days that feeding was withheld for ≥12 h due to gastric residuals greater than 50% of the prior feeding or more than 2 mL/kg, bile- or bloodstained gastric residuals, emesis, abdominal distention or tenderness, changes in stool pattern or consistency, or presence of blood in the stool.

Growth velocity

Growth velocity was calculated in the Assad study [18] to assess weight gain from birth to discharge using the two point average weight GV = [1000 × (Wn − W1)]/{(Dn − D1) × [(W1 + Wn)/2]}. Where W = weight in grams, D = day, 1 = beginning of time interval, and *n* = end of time interval in days. 

### 2.6. Assessment of Methodological Quality

All included studies were assessed for methodological quality. The risk of bias was assessed using elements of the Cochrane Collaboration tool 2.0 for randomized studies [19]. For non-randomized studies, the risk of bias was assessed using a modified Newcastle–Ottawa scale [20]. The following domains were evaluated: selection, comparability and outcome. A score of >7/9 was deemed low risk, a score of 4–6/9 was deemed a moderate risk, and a score of ≤3/9 was deemed a high risk of bias. Two authors (SD, PT) performed the risk of bias independently; conflicts were resolved after discussion and consensus. PT and SD assessed the certainty of evidence (confidence in the estimate of effect) for each outcome based on the Grading of Recommendations Assessment, Development, and Evaluation framework (Table 2). Any discrepancies were resolved by discussion and consensus (BK).

### 2.7. Summary Measures

Four clinical data sets fulfilled the search criteria (Appendix A).

#### 2.7.1. The study by Jensen et al. [14] [NORDIC]

This was the first prospective randomized controlled trial that compared the effects of human milk-based and bovine milk-based nutrient fortifiers, resulting in the exclusively human breast milk diet being more effective. Infants were randomized before the enteral feeds reached 100 mL/kg/day, which was at day 6 of life. The trial was conducted to evaluate, with a combined outcome, the composite of NEC stage II–III, culture-proven sepsis, and mortality from inclusion to discharge (NCT03797157). Information on the proportion of MOM versus pooled DM as basis of the human milk diet was not available. Individual composition of the fortified milk was not available [21].

#### 2.7.2. OptiMoM Trial [15]

This was the first dedicated clinical study to compare HM-HMF to HM-BMF in a blinded, multicenter RCT with 125 infants of <1250 g birthweight (NCT02137473). The primary outcome was feeding interruption, by which the trial was powered. A secondary outcome was a dichotomous mortality/morbidity index, which was affirmative for any one or more out of death, LOS, BPD, ROP, or NEC (Bell’s stage II or greater) [15]. All infants received MOM supplemented by donor milk (DM) if required. Randomization to BMF or HMF happened at the introduction of fortification on day 17 of life. Partial HM fortification was started at a feeding volume of 100 mL/kg per day and full fortification was at 140 mL/kg/day. Information on the proportion of MOM versus pooled DM as the basis of the human milk diet was not available. Individual composition of the fortified milk was not available.

#### 2.7.3. Subgroup Analysis of NCT00506584 [Sullivan trial] from Lucas et al. [16]

An RCT with 3 limbs in 11 centers in the USA and 1 in Austria was the initial trial [22]. Infants with a birthweight of 500–1250 g at birth were included. All infants were fed MOM or standardized pooled DM. The two intervention groups were fed a liquid HMF at 100 mL/kg/day or at 40 mL/kg/day, respectively. The control group received HM-BMF. A subgroup analysis of the trial was performed by Lucas et al., 2020 [16]. They included infants with a 100% mother’s milk diet. With this analysis, the only difference between the subgroups was fortification with HMF or with BMF. The infants fed different concentrations of liquid HMF were merged into one group since there were only small differences. Therefore, a total of 82 patients were on HM-HMF and 32 patients were on HM-BMF. The primary outcome of the original trial was NEC with BPD, ROP, and sepsis as secondary outcomes. Information on the proportion of MOM versus pooled DM as the basis of the human milk diet was not available. Individual composition of the fortified milk was not available.

#### 2.7.4. Subgroup Analysis of NCT00506584 [Assad trial] from Lucas et al. [17]

Assad et al. published a single-center retrospective study [18]. A total of 293 preterm infants were included. The analyses were performed along with a quasi-experimental comparison with enteral therapy based on HM-BMF before 2012 and HM-HMF after 2012. The subgroup analysis required a reanalysis of primary data which was carried out by Lucas et al., 2020 [17]. Information on the proportion of MOM versus pooled DM as the basis of the human milk diet was not available. Individual composition of the fortified milk was not available. 

### 2.8. Data Synthesis and Statistical Analysis

All the studies were combined and analyzed using Cochrane’s Review Manager (RevMan5). For continuous outcomes, the mean difference with 95% CI was calculated, and for dichotomous outcomes, the RR with 95% CI was calculated from the data provided in the studies. Random effects model was used to calculate summary statistics owing to anticipated heterogeneity. Statistical heterogeneity was assessed by using the Cochrane *Q* statistic and the *I*^2^ statistic, which is derived from the *Q* statistic and describes the proportion of total variation that is due to heterogeneity beyond chance.

## 3. Results

### 3.1. Primary Outcomes

#### 3.1.1. Death

Human milk fortification using HMFs reduced mortality by 50% (ARR: 3.2% and NNT: 31) as compared to BMF (467 infants; RR = 0.50, 95% CI = 0.25–0.97; *p* = 0.04; *I*^2^ = 0%) from three RCTs. There was no heterogeneity among the studies included. When RCTs and NRSI were combined the results were similar (four studies, 681 infants; RR = 0.50, 95% CI = 0.26–0.94; *p* = 0.03; *I*^2^ = 0%) favoring HM-HMFs (Figure 2A,B).

#### 3.1.2. NEC

NEC stage II or beyond did not differ significantly (four studies, 681 infants; RR = 0.48, 95% CI = 0.20–1.17; *p* = 0.11; *I*^2^ = 39%) when combined or when analysis was performed separately for RCTs alone (467 infants; RR = 0.61, 95% CI = 0.26–1.42; *p* = 0.25; *I*^2^ = 28%) as shown in Figure 2C,D. Jensen et al. [14], O’Connor et al. [15], and Lucas et al. (Sullivan et al.) [16] all used the modified Bell’s stage for the diagnosis of NEC. The subgroup analysis from the original study by Sullivan et al. [22] did show a significant reduction in NEC (*p* = 0.038)/surgical NEC (*p* = 0.014) in the group of infants with HMFs. 

### 3.2. Secondary Outcomes

#### 3.2.1. BPD

The incidence of BPD did not differ significantly between the two groups (four studies, 663 infants; RR = 0.83; 95% CI = 0.69–1.00; *p* = 0.05, *I*^2^ = 0%) as shown in Figure 3A. When RCTs [14,15,16] were analyzed separately, the effect size of pooled estimate was not significant either (449 infants; RR = 0.86; 95% CI = 0.70–1.04).

#### 3.2.2. Sepsis

Jensen et al. reported data on sepsis [14]. The other three studies reported late-onset sepsis as the outcome [15,16,18]. Pooled data from four studies (*n* = 681) showed no significant differences in sepsis/LOS between the HM-HMF and HM-BMF groups (RR = 0.97, 95% CI = 0.66–1.42; *p* = 0.87; *I*^2^ = 26%; Figure 3B). 

#### 3.2.3. ROP

All included studies [14,15,16,18] reported data on ROP. Three studies analyzed severe or grade 3/4 ROP, while Assad et al. [18] reported on any ROP (Figure 3C). A meta-analysis of the pooled data showed no difference for combined ROP (four studies, 671 infants; RR = 0.64, 95% CI = 0.28–1.44; *p* = 0.28; *I*^2^ = 69%). Assad et al. [18] found less ROP with the HM-HMF diet (214 infants; RR = 0.40, 95% CI = 0.22–0.74; *p* = 0.003; *I*^2^ = 69%).

#### 3.2.4. Feed Intolerance and Growth Velocity

Feed intolerance, as defined by the authors, was reported in two studies [14,15] and was not significantly different between two groups (two studies, 353 infants; OR = 0.76, 95% CI = 0.48–1.19; *p* = 0.94; *I*^2^ = 0%). Growth velocity (g/kg/d) was also not different (two studies, 328 infants; SMD: −0.59 to 0.64; *p* = 0.93; *I*^2^ = 0%) (Figure 3D,E). Information on MOM or pooled DM as the basis of the human milk diet was not available.

#### 3.2.5. Risk of Bias Assessment and GRADE

The risk of bias assessment was performed using the ROB.2 tool (Appendix A). All three included RCTs [14,15,22] were adjudged with ‘some concerns’. Two RCTs did not attempt blinding due to the nature of the interventions.

The certainty of evidence (CoE) is provided in Table 2 and was ‘very low’for the primary outcome of mortality. We downgraded the evidence by three levels due to a risk of bias, indirectness, and imprecision. For the outcome of NEC, the CoE was ‘low’ due to a risk of bias and inconsistency. 

## 4. Discussion

Exclusively using human milk plus human milk-derived fortifiers versus using human milk plus bovine milk-derived fortifiers for extremely low-gestational-period preterm infants is a complex issue. First, bovine milk products have been introduced into neonatal care without a safety consideration or verification [23]. Secondly, the gestational age at which preterm infants could survive has been significantly lowered in the last two decades, further changing the susceptible phase of immune development [24]. Thirdly, clinical studies that have addressed the research question with robust methodology and adequate power are sparse [25]. Therefore, the value of a meta-analysis is high despite there being heterogeneity among the studies. Lastly, human milk is a tissue that is collected according to standards both for donation and processing with strict quality control standards to ensure safety and effectiveness [26], resulting in accepted protocols [27]. However, the implementation of human milk-based fortifiers as a standard must be thoroughly warranted given the high costs [28] in the individual national health care system [29]. 

Our meta-analysis included a total number of 681 infants. Our meta-analysis studied the short-term outcomes of death, BPD, NEC, sepsis, and ROP in all four included clinical data sets. The new trial affected the readouts of the previous meta-analysis considerably. Lucas et al. reported in 2020 that NEC, ROP, PDA, and feed withheld for >24 h were significantly increased in an HM-BMF diet [16]. The most important finding of our analyses was the reduction in mortality across all four clinical studies and data sets. The number needed to treat was 31.The omission of Assad et al. [18], a retrospective study, did not change the findings. Since death was the most important outcome, we cannot discard the subgroup analyses for formal reasons. It is very clear that these findings mandate subsequent clinical trials to confirm it in additional randomized, (if possible) blinded clinical trials. However, some neonatologists will not have a clinical equipoise anymore, which is a prerequisite for such trials. We could not establish the reason for increased mortality in the bovine fortifier group, and can only speculate that it is due to the cumulative effect of the morbidities rather than effect of the bovine fortifier alone. In the HMF-HM group, 14 infants died versus 24 infants in the BMF-HM group. The difference is not significant. The common causes of death in this period of life in preterm infants may be sepsis, NEC, pulmonary complications, or IVH. Additional information about the cause of death of the patients in the studies was not available. Therefore, we are unable to provide any additional analysis on the difference in mortality, but instead have to assume a cumulative effect.

The NEC incidence amongst the four studies reflects the general improvement in outcomes in recent years. The most recent study by Jensen et al. had an incidence of 7.5% [14]. The Sullivan study had the highest incidence of NEC (16%) [16]. The interpretation of these results of the post hoc analysis was not without problems [30]. The current incidence of NEC makes adequately powered clinical trials mandatory, but it is difficult for them reach statistical significance. Future clinical trials need to test if human milk-derived fortifiers decrease the rate of NEC while also taking into account the differences between pooled donor milk and mother’s own milk.

In our analysis, death was significantly reduced, with the possible benefit of reducing BPD with *p* = 0.05. However, considering the observational nature of the majority of the included data set, drawing causality was not appropriate. Nonetheless, the benefits to the survival and to the prevention of BPD were the most consistent associations across all datasets. Reaching the *p*-value of *p* = 0.05 for BPD prevention is challenging. Amrhein et al. [31] questioned the dichotomous use of *p*-values. A *p* value of 0.05 is an arbitrary cut-off which does not justify the determination of “non-significant”, especially in the context of large RRs. A clinical decision on how to proceed is warranted until further clinical evidence becomes available given the enormous importance of survival and BPD prevention [32]. ROP was not different between the two groups. The analysis, however, cannot be considered accurate, since different classifications were used and different outcomes were reported, respectively. Feeding intolerance and growth rates were not different between the two groups either. The fact that the composition of MoM is changed by the addition of any fortifier is good news. Particularly since BMFs are mostly powders which mandate resuspension compared to HMFs which are usually frozen liquids. If the human milk must be warmed for the resuspension of the powder, the shelf life of the preparation is strictly limited. If a frozen liquid is used as a fortifier, mother’s own milk will be replaced in the total volume of the feeding. The importance of these effects has never been studied. The theoretical possible effects may comprise changes in micro- and macronutrient composition, electrolyte concentrations, and caloric content which might result in poorer growth. Since there are no clinical studies available, the possible effects are purely speculative and need to be tested in the future. At least for the analyzed studies, there were no differences concerning growth rate and feeding intolerance that were associated with the studied fortifiers.

The sepsis rate was not different between the groups either. Other factors that play a role in the overall outcome of preterm infants include inborn or out-born birth, expertise of the center, dedication of the medical team to lactation support, parental education, and their socio-economic status, in addition to general hygiene standards to minimize nosocomial infections. Since the infants were randomized per center in all included studies we assume an even distribution of these factors in the two groups. The most straightforward and appropriate solution may be the shared decision making with the parents [33,34]. Decision making on enteral therapy with parents is a very important part of the parents’ experience in the NICU. It should be a continuous process which should start before birth. Given the extreme emotional situation and distress of the parents after a preterm birth, it is imperative to provide information, education, guidance, and support in order to empower them in their parenting role and to treat them as equal partners. The different feeding options available including mother’s own breast milk, pooled donor milk, and the use of HMFs versus BMFs need to be communicated in a timely and understandable manner. There are inherent differences between mom’s own milk (MoM) and pooled donor milk [35]. None of the included studies took this practical aspect into consideration. Unless daily analysis and subsequent fortification is carried out, we cannot reach a standardization. An exclusive human milk diet has nutritional benefits and avoids the possible allergens present in bovine milk (or soy-based) formulas or fortifiers [36]. Parents must understand and agree to the treatment plan voluntarily, including accepting the inherent risks and benefits in shared decision-making. The parents’ preferences and values must be respected in terms of cultural, religious, and personal beliefs that may influence their decision on bovine milk products [37].

### Limitations

Our meta-analysis could not fulfill the criteria of a conventional systematic review or meta-analysis since published evidence for two of the four studies (Sullivan et al. and Assad et al.) were only secondarily published [16]. This was also the reason why the recent Cochrane review only included the OptiMoM trial [25]. Subgroup analyses have the inherent risk of imbalances between the groups. In each of the two studies under consideration, the baseline risk factors were, however, well balanced.

The time of life when BMF was introduced ranged from day 6 to 17 of life between the studies. The adverse effects of bovine milk can be moderated by the later introduction into the diet despite the adverse effects on growth [12]. 

All included studies did not report the quantity of fresh mother’s own milk or pooled donor milk separately. The difficulty in distinguishing the effect of fresh human milk and pooled donor milk in addition to the type of fortifier used is a major limitation of the included studies and thus this meta-analysis. The individual composition of the fortified milk could not be recorded. Lucas et al. estimated the proportion of bovine milk protein in the diet to be 60%, which may be a common feature in all studies despite all other differences [16]. Conflating birthweight and gestational age as inclusion criteria is not ideal; however, this was performed due to a lack of sufficient data from the clinical trials.

The studies were conducted in different years and centers, also reflecting the general progress of perinatology which improved clinical outcomes over the years. Therefore, the magnitude of these general changes in practice cannot be addressed.

Irrespective of how neonatologists, nurses, parents, health insurances, and health departments will use the results of this meta-analysis, the questions about an exclusive human milk diet are not completely answered. An assessment of the benefits of exclusive human milk diet has the traits of a medical reversal: a treatment is introduced for beneficial short-term read-outs which are irrelevant to the important long-term outcomes, as is the case for drugs for cardiac arrythmias [38]. However, we still lack sufficiently powered clinical trials and relevant long-term outcomes in terms of neurodevelopment. Although BPD itself is a disease for life and is associated with poorer neurodevelopmental outcomes, we need neurodevelopmental follow-up data for all survivors to definitely assess if the use of an exclusive human milk diet is warranted for the reduction of relevant outcomes in addition to death and BPD [7].

## 5. Conclusions

Our data associates bovine milk-derived fortifiers with a possibly increased risk of death, which makes a reversal possibly necessary. However, the introduction of bovine milk fortifiers cannot yet be judged due to the lack of sufficiently powered clinical trials and the lack of relevant information about the long-term outcomes in terms of neurodevelopment. Although BPD itself is a disease for life and is associated with poorer neurodevelopmental outcomes, we need neurodevelopmental follow-up data from all survivors to definitely address the question of if the use of an exclusive human milk diet from MOM and/or pooled DM is warranted due to the unique nutritional and immunological benefits from human breast milk which can reduce the relevant outcomes of an extremely low gestational period. The results should not be refuted for formal reasons but should be taken as the need to further define the effects of a human milk diet (MOM and/or pooled DM) supplemented with human milk fortifiers.

## Figures and Tables

**Figure 1 nutrients-16-00910-f001:**
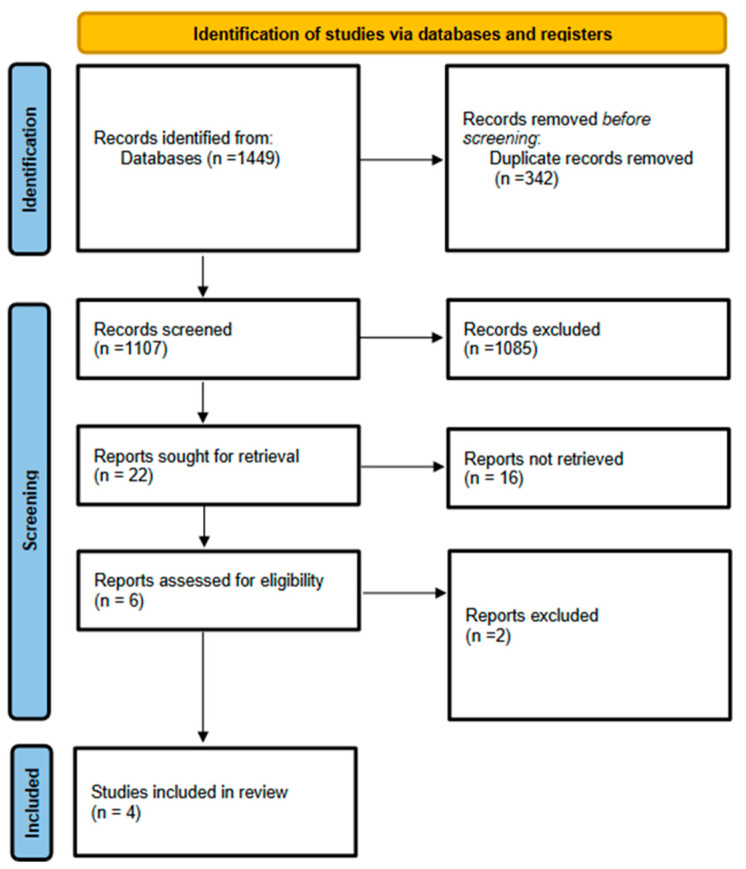
PRISMA flow diagram.

**Figure 2 nutrients-16-00910-f002:**
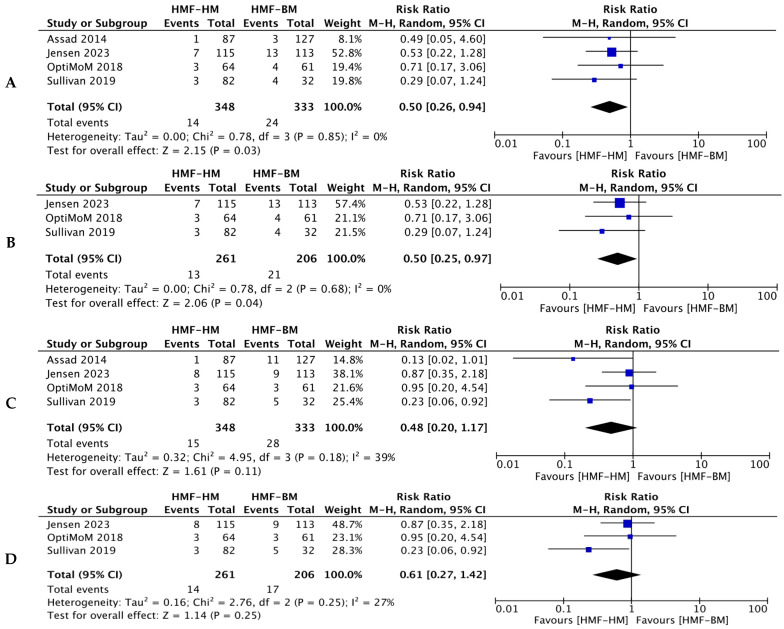
Forest plots for primary outcomes. (**A**). Mortality (all studies, ARR 3.2% NNT-31) [14,15,16,18]. (**B**). Mortality (RCT’s) [14,15,16]. (**C**). NEC stage ≥ 2 (all studies) [14,15,16,18]. (**D**). NEC stage ≥ 2 (RCT’s) [14,15,16].

**Figure 3 nutrients-16-00910-f003:**
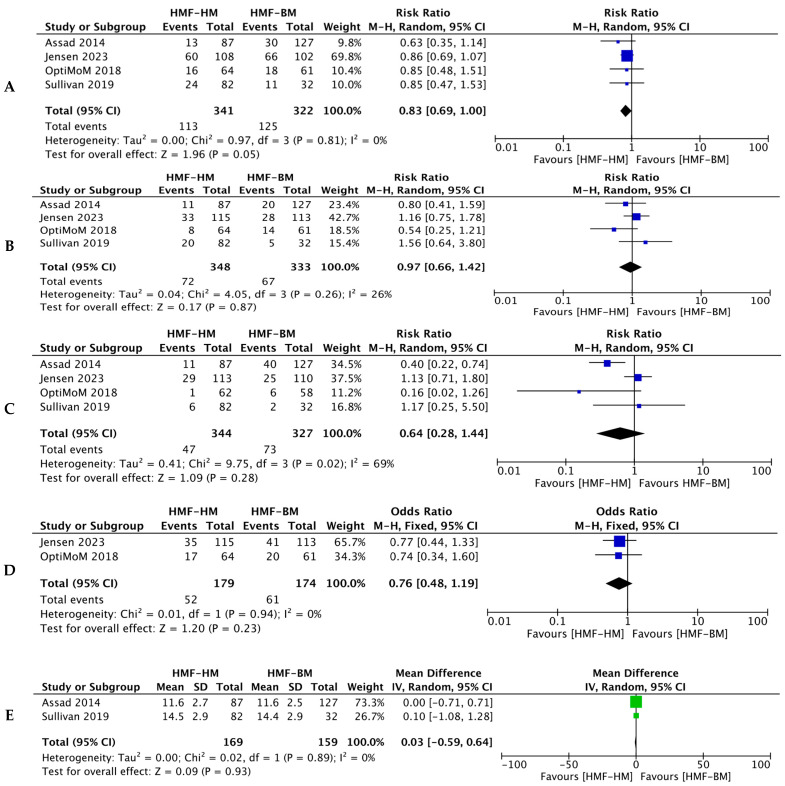
Forest plots for secondary outcomes. (**A**). BPD [14,15,16,18]. (**B**). Late-onset sepsis [14,15,16,18]. (**C**). ROP [14,15,16,18]. (**D**). Feed intolerance [14,15]. (**E**). Growth Velocity [16,18].

**Table 1 nutrients-16-00910-t001:** Summary of all included studies.

Study ID	Study Type	Sample Size (HMF-HM vs. HMF-BM)	GA(wks) or BW (g)	Primary Outcomes	Secondary Outcomes	Formulation HMF and BMF	Fortification Started at	End of the Intervention	Confounding Factors
Jensen, 2023 [14] Sweden	RCT	115 vs. 113	<28 wk	Composite of NEC stage II-III, Culture proven sepsis and mortality	NEC, death, sepsis, BPD, ROP, PVL, intensive care days, mechanical ventilation days, feeding intolerance	Humavant + 6, Prolacta vs. BMF of the responsible unit	100 mL/kg/day	34 wks PMA	NI
O’Connor, 2018 [15] Canada	RCT	64 vs. 63	<1250 g	Feeding interruption	Other measures of feeding tolerance, a dichotomous mortality and morbidity index (death, LOS, BPD, ROP, or NEC), fecal calprotectin, growth	Prolact +4/Prolact +6/Prolact +8, Prolacta vs. Similac Human Milk Fortifier Powder, Abbott Nutrition	100 mL/kg/day	Whichever came first: 84 d of age/discharge/≥2 complete oral feeds daily over 3d	GA
Subgroup analysis of NCT00506584 [Sullivan trial] from Lucas et al., 2020 [16] USA, Austria	RCT	82 vs. 32	<1250 g	NEC	BPD, requirement for mechanical ventilation, ROP, sepsis, growth	Prolact + H2MF, Prolacta vs. Enfamil, Mead Johnson or Similac, Abbott Nutrition	40 or 100 mL/kg/day for HMF and 100 mL/kg/day for BMF	Whichever came first: 91d of age/ discharge/≥4 complete oral feeds/day	GA
Subgroup analysis of NCT00506584 [Assad trial] from Lucas et al., 2020 [17] USA	Retrospective cohort	87 vs. 127	<28 wk or <1500 g	Length of time to full feds, length of stay, incidence of feeding tolerance	NEC, costs	Prolact + H2MF, Prolacta vs. Similac, Abbott Nutrition	120–150 mL/kg/day	Discharge	GA, BW

HMF-HM: human milk with human milk fortifier; HMF-BM: human milk with bovine milk fortifier; GA: gestational age; BW: birth weight; HMF: human milk fortifier; BMF: bovine milk fortifier; RCT: randomized clinical trial; NEC: necrotizing enterocolitis; BPD: bronchopulmonary dysplasia; ROP: retinopathy of prematurity; PVL: periventricular leukomalacia; PMA: postmenstrual age; NI: not identified; LOS: late-onset sepsis.

**Table 2 nutrients-16-00910-t002:** Grading of Recommendations Assessment, Development, and Evaluation (GRADE) framework.

Outcomes	Anticipated Absolute Effects * (95% CI)		Relative Effect (95% CI)	No. of Participants (Studies)	Certainty of the Evidence (GRADE)
	Risk with HMF-BM	Risk with HMF-HM			
Mortality	10 per 100	5 per 100 (3 to 10)	RR 0.50 (0.25 to 0.97)	467 (3 RCTs)	⨁◯◯◯ Very low ^a^
NEC stage ≥ 2	8 per 100	5 per 100 (2 to 12)	RR 0.61 (0.27 to 1.42)	467 (3 RCTs)	⨁⨁◯◯ Low ^b^
BPD at 36 weeks PMA	49 per 100	42 per 100 (34 to 51)	RR 0.86 (0.70 to 1.04)	449 (3 RCTs)	⨁◯◯◯ Very low ^c^
Retinopathy of Prematurity (any)	17 per 100	14 per 100 (5 to 35)	RR 0.84 (0.33 to 2.13)	457 (3 RCTs)	⨁◯◯◯ Very low ^d^
Late-onset Sepsis	23 per 100	23 per 100 (14 to 39)	RR 1.01 (0.60 to 1.71)	467 (3 RCTs)	⨁◯◯◯ Very low ^e^
Summary of findings (SoF) table					
HMF-HM compared to HMF-BM in Extreme Preterm Outcomes					
Patient or population: Extreme preterm or VLBW infants Setting: Neonatal Intensive Care Unit Intervention: HMF-HM Comparison: HMF-BM					

Explanations. * The risk in the intervention group (and its 95% confidence interval) is based on the assumed risk in the comparison group and the relative effect of the intervention (and its 95% CI).^a^. We downgraded the evidence by three levels due to lack of blinding, indirectness, and inconsistency. ^b^. We downgraded the evidence by two levels due to lack of blinding in two studies and inconsistency or lack of blinding. ^c^. We downgraded the evidence by three levels due to lack of blinding, indirectness, inconsistency, and imprecision. ^d^. We downgraded the evidence by three levels due to lack of blinding, indirectness, inconsistency, and imprecision. ^e^. We downgraded the evidence by three levels due to lack of blinding, indirectness, inconsistency, and imprecision.

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
