# Peer review of "Association of Fortification with Human Milk versus Bovine Milk-Based Fortifiers on Short-Term Outcomes in Preterm Infants—A Meta-Analysis"

_nutrients, 2024, doi:10.3390/nu16060910_

Round 1

Reviewer 1 Report

Comments and Suggestions for Authors

It is a well constructed meta-analysis on human milk versus bovine milk based fortifiers.

The design of the study was well structured, and the analysis of the results was also well conducted.

Please make a few minor corrections.

1. Significant result related to death was confirmed. The authors expressed the reason unclearly, but it seems like more analysis is needed.

2. Additionally, it seems necessary to add discussion on factors other than death and BPD.

Reviewer 2 Report

Comments and Suggestions for Authors

Overall, this paper is well-written and provides clinically relevant information

Comments on the Quality of English Language

There are some grammar and spacing issues; please revise appropriately.

Introduction

  1. I disagree that we have “shifted from reducing mortality to reducing morbidity.”  VLBWs still experience a high mortality. 
  2. This sentence is misleading “Bovine milk products have been associated with increased rate of NEC depending on the 55 time of introduction into enteral therapy[12, 13].”  These studies show formula, not bovine-milk fortifiers, are associated with NEC.
  3. Do you think the displacement of MOM with HMF is problematic?  Any concern about electrolyte derangements with HMFs, poor growth, need for “higher” fortification?  

Methods

  1. I would define sepsis, LOS, BPD, and feeding intolerance and growth velocity in the methods.  
